# The Effects of China’s Targeted Poverty Alleviation Policy on the Health and Health Equity of Rural Poor Residents: Evidence from Shaanxi Province

**DOI:** 10.3390/healthcare8030256

**Published:** 2020-08-06

**Authors:** Xiuliang Dai, Lijian Wang, Yangling Ren

**Affiliations:** School of Public Policy and Administration, Xi’an Jiaotong University, Xi’an 710049, China; daixiuliang@163.com (X.D.); ryling0317@163.com (Y.R.)

**Keywords:** China’s targeted poverty alleviation policy, rural poor residents, health equity, concentration index, decomposition of the concentration index

## Abstract

*Objective*: China’s targeted poverty alleviation policy has a profound impact on the country’s rural economic and social development now. This study aimed to learn about the health status and health equity of rural poor residents under the implementation of the policy. It further explores the factors affecting the health status and health equity of rural poor residents in order to contribute to the improvement of the policy. *Methods*: The data of 1233 rural poor residents were collected from a questionnaire survey from 12 prefecture-level cities and areas of Shaanxi province in 2017, and the self-reported health was used to reflect the health status. A concentration index was applied to measure the inequity of the health status of rural poor residents. The decomposition method was employed to explore the source of health inequity. Results: The results showed that 44.56% of rural poor residents in Shaanxi province had a poor or very poor health status, which was affected by their economic level, gender, age, degree of education, and marital status. Additionally, participation in agricultural industry development, relocation, health poverty alleviation, and basic living standards were significantly correlated with health status. The concentration index of the health status of rural poor residents was 0.0327. The primary contributors to the health inequity in different regions varied, but the economic level and the degree of education were the most significant factors, and the targeted poverty alleviation policy had a significant impact on health equity. *Conclusions*: The results indicated that the health status of rural poor residents in Shaanxi province was generally poor, there was a pro-rich inequity in the health status, and the degree of education and economic level were the primary factors affecting the health status and health equity. The targeted poverty alleviation policy greatly impacted the health status and health equity of rural poor residents, and the difference in health status would lead to the inequity of benefits of the targeted poverty alleviation policy. In the future, the policy should focus on ensuring the sustainable development ability of rural residents with poor health status.

## 1. Introduction

Health is not only a direct component of the well-being of humans but also a human capital that increases the development capacity of individuals, families, and society [1]. There is a close relationship between health and poverty, and poor residents with a poor health status due to the lack of health investment and access to health care lead to the exacerbation of their own poverty. Previous studies have shown that lower socioeconomic levels tend to reduce residents’ enthusiasm to invest in health, and groups with lower socioeconomic status tend to face higher health risks [2]. Therefore, the development of public health through national systems and policies plays an important role in promoting individual health and eliminating poverty [3]. At the same time, ensuring the health equity of different social and economic status groups has always been an important goal of medical and health system reforms in various countries. Health equity is an important part of social equity and justice, which is the basis of ensuring that different groups have equal access to resources and viability [4]. By the end of 2012, there were 98.99 million poor people in China’s rural areas. Another important fact is that poverty due to disease is an important cause of the poverty of rural residents in China. Over time, there has been a great difference between urban and rural economic development and insufficient investment in rural public health services, leading to health problems and poverty within rural residents in China [5]. So, for a long time, the Chinese government has made great efforts to promote the health of citizens and eliminate poverty. Since 2013, China’s targeted poverty alleviation strategy has become an important tool and source of power to help rural poor residents shake off poverty and build a moderately well-rounded prosperous society.

Targeted poverty alleviation means to promote the income, living standards, and health of rural poor residents through comprehensive poverty reduction measures. “The 13th five-year plan for poverty alleviation (2016)” has set the development goals of “ensuring that the rural poor have enough food and clothing under the current standards, and that compulsory education, basic medical care, and housing are secure”. We formulated a targeted poverty alleviation policy system, consisting of developing competitive industries, locating jobs elsewhere, relocation, improving education, providing better healthcare, better ecological protection, guaranteeing basic living standards, and social poverty alleviation, which is listed in Table 1. Rural poor residents refer to the poverty-stricken family members that meet the standards of poor households in the targeted poverty alleviation policy and have been accurately identified by the government. The standards of poor households are the per-capital net income less than 3050 yuan in 2013 (1 yuan ≈ 0.1621 dollars), and food, clothing, compulsory education, basic health service, and safe housing are not ensured. According to a report by China’s national bureau of statistics, by the end of 2018, the number of people living in poverty in rural China had dropped from 98.99 million at the end of 2012 to 16.6 million, and the poverty rate had dropped from 10.2% to 1.7%. The targeted poverty alleviation strategy plays a significant role in promoting economic development, environmental improvement, social progress, and the improvement of people’s health status in rural poverty-stricken areas. The targeted poverty alleviation policy not only increases the income of rural poor residents but also reduces their medical burden, which has a profound impact on improving their health status. But due to the influence of personal and family development ability, the policy benefits will differ between residents. Ensuring the fairness of benefits for poor residents is also an important principle for the implementation of the targeted poverty alleviation policy; therefore, it is important to focus on the health and health equity of the rural poor residents.

Although previous research has paid some attention to China’s targeted poverty alleviation policies, the health and health equity of the rural poor residents under the policy have not been addressed. The previous research has focused on the following aspects. First, it explores relevant policy innovations, such as land policies, in the implementation process of the targeted poverty alleviation strategy [6,7]. Second, the implementation effect of the targeted poverty alleviation policy is evaluated based on different perspectives and methods [8,9,10,11]. Third, it analyzes the challenges and innovation paths of the implementation of the targeted poverty alleviation policy [12,13]. Meanwhile, researchers have made full research on China’s rural medical policies and effects [14,15,16]. But there is still a lot of research gap on the health and health equity of poor rural residents in China under the background of targeted poverty alleviation policy. This important problem has not attracted enough attention from researchers. Therefore, this research has explored the effect of the policy on the health and health equity, which is conducive to improving the implementation of the targeted poverty alleviation and promoting the health status and health equity of rural poor residents. 

## 2. Materials and Methods

### 2.1. Data Collection and Quality Control

#### 2.1.1. Organization and Implementation

In economics, the panel data was often used to assess the effect of external factors on self-assessed health [17]. Because there was no panel data for us to use, this study used the data collected in 2017 from Shaanxi province to describe and analyze the health status and health equity of rural poor residents. Shaanxi province is a key region for the implementation of China’s targeted poverty alleviation strategy. By the end of 2018, there were still 29 poverty-stricken counties with 775,500 poor residents. Affected by the natural environment, Shaanxi province is divided into three regions: Shanbei, Guanzhong, and Shannan. Shanbei is the loess plateau, Guanzhong is the plain, and Shannan is the mountainous area, and the economic and social development levels of the three regions are quite different. So, it is also necessary to analyze the health status and health equity of rural poor residents from the perspective of regional differences. The identification standards for poor residents in Shaanxi province include: first, the net per capita annual income of residents is less than 3070 yuan; Second, food and clothing (including safe drinking water) are not guaranteed, and compulsory education, basic medical care, and housing security have not been effectively addressed.

The research team carried out a questionnaire survey on rural poor residents in Shaanxi province in 2017. The questionnaire included basic information of rural poor residents, their living conditions, their health status, and their participation in the targeted poverty alleviation policy. We recruited students who participated in the summer holiday social practice program for college students of authors institute to be the surveyors, which is organized and implemented by the university. The university, together with the governments of 12 prefecture-level cities of Shaanxi province, has established social practice bases for college students. Every summer holiday, the students go to counties (districts) or towns to participate in social practice, which mainly includes working in the government as an intern and carrying out social surveys.

#### 2.1.2. Sampling Method

First, a three-stage sampling survey method combining probability and non-probability sampling was adopted. We selected two counties (districts) with rural areas from each prefecture-live city in Shaanxi province, and two typical rural administrative villages were selected in each county (district). Last, each rural village randomly selected twenty eligible poor residents as the subject of the survey. The rural poor residents were selected from the list provided by the government. The sampling process was carried out by each survey team in accordance with scientific, typical, and convenient principles.

#### 2.1.3. Data Quality Control

The surveyors who participated in the survey were trained ahead of time. The structure and content of the questionnaire were explained, and the matters that required attention and principles in the questionnaire survey were emphasized. Every member of this survey could use the survey data to conduct research. Besides, problems existing in the survey were solved timely through network communication during the implementation of the survey. Last, the surveyors checked, inputted, and cleaned the returned valid questionnaires to ensure data quality. A total of 1233 valid samples were obtained after removing the samples under the age of 16.

### 2.2. The Methods of Data Analysis

#### 2.2.1. Description of the Health Status of Rural Poor Residents

Descriptive statistical analysis was used to describe the health status of rural poor residents, and the chi-square test was used to analyze whether there were statistically significant differences in the health status of poor residents in different regions.

#### 2.2.2. Measurement of Health Equity

A simple way to measure health equity is to test whether two groups (the poor and the rich) have the same health level. Currently, there are many methods to measure health equity, including the method of concentration curve and concentration index, method of Lorenz curve and Gini-coefficient Lorenz curve, Atkinson index, and chi-square value method [18]. Although the method of concentration index and Lorenz curve and Gini coefficient is similar to the way of expression. The concentration index not only provides an indicator of health inequity but can also be decomposed proportionally into contributions of different inequity of health determinants [19]. Referring to existing studies on health equity of Chinese residents [20,21], this research used the centralized index method to measure the health equity of rural poor residents. The concentration index was used to investigate the inequity degree of a certain variable associated with social and economic status, which dynamically reflects the effect of the variable influenced by income [19]. The concentration index was calculated using Equation (1) [20].
(1)$$C=\frac{2}{u}\mathrm{cov}(y,r)$$
where *C* is concentration index, *y* is health status, u is the mean of health status, r is the fractional rank of income, ranging from 0 to 1. The value of the concentration index was −1 to 1. If the concentration index was 0, this showed that rural poor residents with different economic levels had the same health status. A positive concentration index indicated that people with higher incomes were healthier than those with lower incomes. Conversely, a negative concentration index indicated that people with lower incomes were healthier than those with higher incomes.

The method of decomposition of the concentration index was used to analyze the contributions of various determinants of health to the inequity in health status. The decomposition of the concentration index is proposed by Wagstaff, which is a straightforward way to decompose the measured degree of inequity into the contributions of various explanatory factors [22]. The positive value of contribution means that the variable contributes to pro-rich inequity, that is, richer individuals have a better health status than the poor, and vice versa [19]. We used the ordinary least squares (OLS) linear regression to decompose the health inequity of rural poor residents because the health status of rural poor residents is a count variable. First, a regression model should be given as Equation (2).
(2)$$y_{i}=\alpha +\beta_{m}x^{m}+\sum^{\;}n\beta_{n}x_{i}^{n}+\sum^{\;}n\beta_{p}x_{i}^{p}+\varepsilon_{i}$$
where $y_{i}$ is the health status; $x^{m}$ is income; $x^{n}$ are need variables; $x^{p}$ are other variables; $\beta_{m}$, $\beta_{n}$, and $\beta_{p}$ are coefficients; $\varepsilon_{i}$ is the implied error term, which includes approximation errors. Then, the concentration index for y can be written as Equation (3):(3)$$\hat{C}=\left(\beta_{m}{\bar{x}}_{m}/\bar{y}\right){\hat{C}}_{m}+\sum^{\;}n\left(\beta_{n}{\overset{¯}{x}}_{n}/\overset{¯}{y}\right){\hat{C}}_{n}+\sum^{\;}p\left(\beta_{p}{\overset{¯}{x}}_{p}/\overset{¯}{y}\right){\hat{C}}_{p}+GC_{\varepsilon }/\overset{¯}{y}$$
where C is the concentration index of health status, and $C_{m}$, $C_{n}$, and $C_{p}$ are the concentration indexes of, $x_{m}$,$x_{n}$ and $x_{p}$. The terms on the right side of Equation (3) denotes the contributions of income, need variables, other variables, and the implied error to inequity.

### 2.3. Variables

#### 2.3.1. Outcome Variables

Health is a complex concept, so there are many methods and indicators to measure the health status of residents. The European five-dimensional health scale (EQ-5D) is widely used by researchers due to its simplicity and high credibility [23,24,25]. Zhang divided the health of the elderly into three aspects: physical health, cognitive function, and self-evaluated health [26]. Li measured the health status of rural Chinese residents by whether they had been ill in the past two weeks and whether they suffered from chronic diseases [27]. Meanwhile, Lorraine et al. thought that self-rated health is generally accepted as a valid measure of health status in population studies in their research [28], and Hesketh et al. used the self-rated health to explore the health status and access to healthcare of migrant workers in China [29]. Considering the use of self-reported health to measure the health status of people for reference, this study thought that self-reported health status could accurately and directly reflect rural poor residents’ understanding of their overall health status, including both physical health and mental health. Therefore, this study reflected the health status of rural poor residents through self-reported health questionnaires. The self-evaluation question about health is “how do you feel about your physical health?”, and the answer includes five dimensions of very poor, poor, average, good, and very good, with a value of 1–5.

#### 2.3.2. Independent Variables

Since this study used the method of centralized exponential decomposition to analyze the influencing factors of health equity, the independent variables in this study included three categories: income, need variables, and other variables. Income is measured by self-reported annual household income. Due to the process of targeted poverty alleviation, the financial sources of poor households may be diverse. In the process of the survey, the surveyor specifically asks whether there are government subsidies and other policy incomes to ensure the accuracy of income. Need variables are closely related to the definition of health equity. In this study, need variables included gender, age, education level, and marital status of rural poor residents. Other variables mainly refer to the targeted poverty alleviation policies that may have an impact on the health and health equity of poor residents. Taking into account the impact of policies and the participation situation of poor residents, the targeted poverty alleviation policies researched in this study included agricultural industry development, relocation, employment help, health poverty alleviation project, and basic living standard guaranteeing. The rural poor residents participated in the agricultural industry development, meaning they developed some agricultural projects with the help of the government, such as agricultural cultivation, livestock breeding, tourism, and E-commerce. The variable descriptions are shown in Table 2.

## 3. Results

### 3.1. General Characteristics of Rural Poor Residents

In the process of social survey, we carried out strict data quality control to ensure the authenticity and scientific nature of the collected data. We used STATA (StataCorp, College Station, TX, USA) for data analysis to ensure the accuracy of data results. So, the external validity of the results was reliable. Table 2 shows the socio-demographic characteristics of the sample and the participation of rural poor residents in Shaanxi province using the targeted poverty alleviation policy. We surveyed 1233 rural poor residents over the age of 15 in Shaanxi province, among whom 74.13% were males, and 25.87% were females. In terms of age distribution, those aged between 16 and 30 accounted for 5.60%, those aged between 31 and 60 accounted for 65.37%, and those aged over 60 accounted for 29.03%. In the education category, the degree of primary school had the largest proportion of 36.10%, followed by rural poor residents who had never attended school at 31.99%, and finally, those with degrees from junior high school or above accounted for 31.91%. In terms of marital status, 12.44% of the rural poor residents were unmarried, and 17.66% were divorced or widowed. Among residents, 14.98% were said to be in very poor health status, 29.58% were in poor health status, 32.56% were in medium health status, and those with good or very good health status accounted for 23.78%. The average annual household income of rural poor residents was about 13,395.11 ± 374.99 yuan (1 yuan ≈ 0.1451 dollars in 2017). According to the participation of rural poor residents in the targeted poverty alleviation policy, 63.16% had participated in the industry development, 17.79% had enjoyed the relocation poverty alleviation policy, 18.68% had enjoyed the employment helping policy, 73.57% had enjoyed the health poverty alleviation policy, and those who enjoyed the basic living standard guaranteeing policy accounted 55.82%. Of the residents, everyone was entitled to enjoy one or more of these targeted poverty alleviation policies.

### 3.2. Comparison of Health Status of Rural Poor Residents under Different Socio-Demographic Characteristics

Table 3 shows the health status of rural poor residents in Shaanxi province under the circumstances of different regions, income levels, gender, age, education level, marriage, and participation in the targeted poverty alleviation, where the chi-squared statistic method was used to test the health status differences. The health status of rural poor residents in varying regions was significantly different. The proportion of rural poor residents with good health status and very good health status in Shanbei was significantly higher than that in Shannan and Guanzhong. Under different family income levels, the health status of rural poor residents had significant differences, the proportion of poor income status with poor health status was 17.28%, the proportion of medium-income status with poor health status was 16.31%, and the proportion of good income status with poor health status was 9.82%. There was a significant difference between male and female health status, the proportion of male residents with general health status or above was 55.06%, while the corresponding proportion of female residents was 59.69%. The health status of poor residents in different age groups was significantly different. The proportion of the poor health status of rural poor residents over 60 years old was 62.54%, which was significantly higher than that of rural poor residents under 60 years old. Rural poor residents with different education levels had different health conditions. The health status of rural poor residents with a junior high school education or above was significantly better than that of those without an education or primary school education. Rural poor residents who were married and cohabiting were in better health status than those who were unmarried, divorced, or widowed.

Judging from the participation of rural poor residents from the targeted poverty alleviation policies, there was a significant correlation between participation in industry development and their health status. The health status of residents participating in agricultural industry development was significantly better than that of residents not participating in agricultural industry development. The proportion of those who participated in the relocation with good health status and very good health status was 25.14% and 2.23%, while the corresponding proportion of those who did not participate in the relocation was 21.55% and 1.79%. There was no significant relationship between participation in employment help and rural poor residents’ health status. There was a significant difference between the poor residents who enjoyed healthy poverty alleviation and those who did not, and the proportion of poor health of the former was 47.41%, which was significantly higher than the latter’s 31.21%. There was a significant difference between the health status of rural poor residents who enjoyed the basic living standard guaranteeing policy and those who did not, a total of 51.89% of rural poor residents who were entitled to the basic living standard guaranteeing policy had very poor health, while the corresponding proportion of the poor residents who were not entitled was 31.17%. These results showed that there was a close relationship between the health status of rural poor residents and poverty and participation in poverty alleviation policies. Industry development and relocation are developmental policies from which healthier residents are more likely to benefit. However, health poverty alleviation and basic living standard guaranteeing are welfare-oriented policies, and the effect on poor residents with poor health is more obvious.

### 3.3. Concentration Index and Decomposition of Health Inequity of Rural Poor Residents

The concentration index of the health status of rural poor residents in Shaanxi province and different regions is shown in Table 4. For the entire sample, according to the concentration index of the health status, there were inequities, favoring those who were richer in Shanxi province (0.0327), and the concentration index of the health status of rural poor residents in Shanbei (0.0068), Guanzhong (0.0593), and Shannan (0.0354) was all positive, meaning there was a health status inequity sloped towards the richer. This showed that residents with a higher income status were more likely to have better health. In addition, compared with the concentration index of rural poor residents’ health status in Guanzhong and Shannan, the concentration index of rural poor residents’ health status in Shanbei was the smallest, meaning that the health equity of rural poor residents in Shanbei was the most advanced.

The decomposition of the health inequity of the rural poor residents is shown in Table 5. The known variables contributed 120.3169% to the inequity of the health status of rural poor residents, and the residual contribution rate was 59.2570%. Among the variables, the most significant contributors to health status inequity of rural poor residents were the degree of education, region, marital status, and age. The degree of education had the highest contribution rate of 32.5702% and a contribution value of 0.0106, indicating that the degree of education increased the inequity of health status, which sloped towards the richer. The contribution rate of the region to the inequity of the health status of rural poor residents in Shaanxi province was 24.3803%, indicating that differences lied in the equity of the health status of rural poor residents from different regions. The contribution rate of marital status, age, economic level, and gender to the inequity of health status was positive, indicating that it increased the inequity of the health status of rural poor residents, which was sloped towards the richer. In terms of income status, the contribution rate of 2.6281% indicated that residents with a higher family income status had better health status. Different targeted poverty alleviation policies had different impacts on the equity of rural poor residents’ health status. The contribution rate of the agricultural industry development, employment help, health poverty alleviation, and basic living standard guaranteeing to health status inequity was 6.8580%, 1.7984%, 10.7070%, and 0.2016%, respectively, which suggested that these policies might do more to improve the health status of the richer residents. However, the contribution rate of relocation policy to the inequity of health status was −3.8396%, which indicated that the relocation policy was more conducive to improving the health status of poorer residents.

The decomposition results of the health status concentration index of rural poor residents in the different regions of Shaanxi province, Shanbei, Guanzhong, and Shannan, are shown in Table 6. In Shanbei, the most significant contributors to health status inequity of rural poor residents were the relocation policy (−121.1139), the degree of education (115.6037%), and economic level (114.2978%), which suggested that the degree of education increased the inequity of health status, which sloped towards the richer. The residents with a higher economic level were more likely to be in better health status, and relocation policy was more conducive to improving the health status of poorer residents. The contribution rate of agricultural industry development (−5.9095%), employment help (−1.4973%), health poverty alleviation (−36.6318%), and basic living standard guaranteeing (−76.1106%) to the inequity of health status of rural poor residents was all negative. In Guanzhong, the most significant contributors to the inequity of the health status of rural poor residents were age (16.4410%), economic level (14.2025%), and the basic living standard guaranteeing (12.3163%). In Shannan, the most significant contributors to the inequity of the health status of rural poor residents were the degree of education (26.5094%), economic level (16.8537%), and the basic living standard guaranteeing (16.0547%).

By comparing the contribution rates of different variables to the inequity of the health status of rural poor residents in different areas of Shaanxi province, it can be found that although the most significant factors in different regions were not identical, economic level and the degree of education were both important factors affecting the inequity of health status, both having positive contribution rates. It was indicated that the rural poor residents with a higher economic level were likely to be in better health status, and the degree of education increased the pro-rich inequity of health status. The effects of the targeted poverty alleviation policy on the inequity of the health status of rural poor residents in different regions of Shaanxi province were different.

## 4. Discussion

We used the sampling survey data of rural poor residents in Shaanxi province in 2017 and found that while the economic level of rural poor residents was low (mean = 13,395.11 ± 374.99 yuan), the overall health status of residents was poor, and the proportion of poor or very poor health status by a self-evaluation was as high as 44.56%. Through chi-square test analysis, it was found that there were significant differences in the health status of rural poor residents in different regions of Shaanxi province, and there were significant correlations between economic level, the degree of education, marital status, age, and health status. There were significant differences in the health status of rural poor residents who participated in industry development, relocation, enjoyed the health poverty alleviation policy, and basic living standard guaranteeing policy. According to the calculations of the concentration index of rural poor residents’ health status and its decomposition, we found that there was an inequity sloped towards the richer for health status. The economic level and the degree of education were the most important factors influencing the inequity of health status, and the targeted poverty alleviation policy had an important influence on the inequity of the health status of rural poor residents.

The research found that the degree of education of rural poor residents in Shaanxi was very low. In terms of the overall situation of Shaanxi province, 31.99% of rural poor residents had no schooling, and just 5% of rural poor residents had a high school education degree or above. The proportion of rural poor residents in Shanbei who had no schooling was as high as 64.58%. The degree of education is an important component of residents’ individual development ability. In the framework of the family sustainable livelihood analysis, education is an important component of livelihood capital [30], and it has a profound impact on the livelihood choices of rural families and their living conditions. A transnational study showed that education is a leading factor of income inequity [31], and poverty has a negative impact on the quantity and quality of education development, which can lead to poverty [32]. The research results showed that poor residents with a lower degree of education were likely to have a poorer health status, and the association was significant in age groups 30 < Age < 61 and Age > 60. For a long time, the relationship between education and health has been highly concerned by researchers [33,34]. Receiving a higher degree of education has a positive impact on the healthy lifestyle and utilization of healthcare of individuals, thus affecting individual health status. Although the current evaluation and identification standard of rural poor residents in China involves a comprehensive target, family income is still an important factor affecting the development of other aspects. There is a complex and close relationship between income, health, and the degree of education. Education and health play a critical role in poverty reduction [35], which shows that a higher economic level is conducive to better education investment and health level.

It was found that the marital status of rural poor residents in Shaanxi province had a significant impact on their health status, and the health status of married cohabitation residents was significantly better than that of unmarried, divorced, or widowed residents. It has been found that marital status influences individual health status through mediating variables, such as social-psychological stress [36]. A study from South Korea added to the evidence that an individual’s health behavior and disease status are linked to marital status [37]. In China’s social context, the family economic level is closely related to men’s marital status, and adult men from poor families have a weaker position in the marriage market. According to the definition of forced male bachelors in rural areas [38], unmarried men aged 28 and above in rural areas are called forced male bachelors. The marriage squeeze has a significant negative impact on men’s quality of life [39]. The forced male bachelors in our survey accounted for 11.31% of all the rural poor male residents, and the proportion of these men with poor health status was as high as 67%. Therefore, we can come to the conclusion that there is also a close link between poverty, marriage, and health status.

There was an important finding, showing a significant relationship between participation in the targeted poverty alleviation programs and the health status of rural poor residents. First, according to the results of the chi-square test, the health status of the poor residents who participated in agricultural industry development was significantly better than that of the poor residents who did not participate in industry development. After controlling the influence of age on health status and participation in agricultural industry development, the difference was still significant. There are two ways to explain this result. One is that poor residents increase their household income by taking part in industry development, which is conducive to improving their health status; another is that poor health status is not conducive to poor residents’ participation in industry development. Participation in relocation has a similar impact on the health of poor residents, as does the participation in industry development. Therefore, it can be concluded that poor residents with poor health status have difficulties in achieving poverty alleviation by participating in industrial development and relocation, and poor health status has become an important factor restricting the development of individuals and families. Second, data analysis results showed that the health status of rural poor residents who enjoyed the health poverty alleviation project and basic living standard guaranteeing project might be worse off. This is because the residents with a poor health status have a higher utilization rate of medical services and medical expenses, and the majority of the poor residents who enjoy the basic living standard guaranteeing project are those who lack the ability to work. In turn, this reflects the support and protection of the targeted poverty alleviation policy to the weak and also reflects the “targeted” of the policy design.

Another issue of great importance is that under the background of implementing the targeted poverty alleviation policies, there was an inequity sloped towards the richer in the health status of rural poor residents in Shaanxi province and different regions. Leading that the higher the family economic level is, the better the health status of the residents may be. Although the Chinese government has long been committed to promoting health equity of different groups through the reform of the medical security system, economic level inequity has always been an important factor affecting the equity of the utilization of health services and the health status of residents [22]. Although the health poverty alleviation project is conducive to reducing the medical burden of rural poor residents, it is still based on the new rural cooperative medical system, which cannot completely eliminate the impact of the family economic level on the equity of the utilization of healthcare services for rural poor residents [40]. Economic level and the degree of education are the most important factors affecting the equity of the health status of rural poor residents, and the degree of education is also an important reason for personal development and family income. The results of the decomposition of the health status concentration index of rural poor residents in Shaanxi province showed that the policies of industry development, employment help, health poverty alleviation, and basic living standard guaranteeing all increased the inequity of the health status sloped towards the richer. At the same time, the effect of the poverty alleviation policy on the equity of the health status of rural poor residents in Shanbei was different from Guanzhong and Shannan. The variables of industry development, relocation, employment help, health poverty alleviation, and basic living standard guaranteeing projects all increased the inequity of the health status sloped towards the poorer. Due to the rural resident’s income gap being higher in Shanbei and the targeted poverty alleviation policies having a more significant effect on the income, there was an increase in the health improvement of residents with a poor economic level.

Above all, it can be concluded that although the targeted poverty alleviation policy promotes the income increase of rural poor residents through industrial development and other measures, improves the health through poverty alleviation, and implements the basic living standard guaranteeing project, the benefits from these policies will impact different people in unequal ways, and the sustainable development of poor residents will also be different. Specifically, improving the ability of the sustainable development of the rural poor residents is the long-term goal of the targeted poverty alleviation policy, and the development-oriented policies, such as industrial development and employment help, are conducive to enhancing the development ability of poor residents by increasing their incomes and through other approaches. However, residents with poor health status are in a weak position during the participation process of these policies, and the positive impact of these policies, in turn, is weakened. An effective measure to ensure the sustainable development ability of residents in poor health conditions is an important issue that needs to be considered to improve the effectiveness of the targeted poverty alleviation policies in the future.

Of cause, there are some limitations to this research. Because of the limitation of the data, this paper mainly discussed the effect of targeted poverty alleviation policy on the health and health equity of rural poor residents based on the analysis of section-cross data and the decomposition of the concentrated index, which is the limitation of this research.

## 5. Conclusions

This research highlighted a rare concern and discussion on the health and health equity of rural poor residents in the context of China’s targeted poverty alleviation policy. Additionally, factors affecting the health status and health equity of rural poor residents were studied by using first-hand survey data, and the research conclusion could significantly improve the effect and equity of the targeted poverty alleviation policy.

This study showed that the rural poor residents in Shaanxi province had a poor health status and low degree of education, with a high proportion of forced male bachelors. The health status of rural poor residents was significantly correlated with family economic level, the degree of education, and marital status, and whether to participate in agricultural industry development, relocation, health poverty alleviation, and utilizing the standard guaranteeing project. The research found that there was an inequity sloped towards the richer on the health status of rural poor residents, the family economic level, and the degree education level, which were the most important factors affecting the health status of poor residents. The economic income of rural residents in different areas of Shaanxi province is significantly different, so policy development should pay more attention to promoting regional equity. The targeted poverty alleviation policy had an important impact on the health equity of rural poor residents. Healthy residents are more likely to participate in poverty alleviation programs to increase their income, while those who are in poor health status are more likely to be trapped in development difficulties. Under the current targeted poverty alleviation policy system, poor residents with poor health status are faced with the double dilemma of increasing family economic income and improving health status, which is shown as the key needed to consolidate and improve the targeted poverty alleviation effect. Therefore, through more scientific and rational policy design, the economic level and health status of poor residents with poor health status should be developed simultaneously, and the equity of policies benefiting different poor residents should be guaranteed, so as to prevent the occurrence of new relative poverty.

There are some important research spaces in this research. First, we would like to try to carry out follow-up research in the future to research the sustainable development of rural poor residents who participated in the targeted poverty alleviation policy. The second is to focus on rural poor residents with poor health status and explore better support policies from the dual perspectives of poverty alleviation and health policy.

## Figures and Tables

**Table 1 healthcare-08-00256-t001:** Main policy systems and measures of poverty alleviation policy in the 13th five-year plan.

Policy Systems	Policy Measures
Agricultural industries (agricultural industry development)	Agricultural cultivation; Livestock breeding; Tourism; E-commerce
Finding jobs elsewhere (employment helping)	Professional training; Transfer of employment; Stabilization of employment
Relocation	Accurately identifying relocation objects; Safe implementation of the relocation
Improving education	Improving the quality of basic education; Reducing the educational burden on poor families; Accelerating the development of vocational education; Improving the ability of higher education to alleviate poverty
Providing better healthcare (health poverty alleviation project)	Improving the ability to provide medical and health services; Enhancing the medical insurance level; Strengthening disease prevention and control and public health
Better ecological protection	Intensifying ecological protection and restoration; Establishing and improving the ecological protection compensation mechanism
Guaranteeing basic living standards	Improving the social assistance system; Gradually raising the level of basic pension in poor areas; Improve the care and service system for the “three left-behind” people and the disabled
Social poverty alleviation	Cooperation on poverty alleviation between the East and the West; The support of Enterprise; The support of army; The support of social organizations and volunteers

**Table 2 healthcare-08-00256-t002:** Socio-demographic characteristics of rural poor residents and variables descriptions.

Variables	Shannxi Province (*N* = 1233) *n* (%)	Shanbei (*N* = 240) *n* (%)	Guanzhong (*N* = 664) *n* (%)	Shannan (*N* = 329) *n* (%)	*p*
Health status					
Very poor	170 (14.98)	12 (5.00)	110 (17.13)	48 (14.77)	0
Poor	357 (29.58)	31 (12.92)	213 (33.18)	113 (34.77)	
Medium	393 (32.56)	131 (54.58)	164 (25.55)	98 (30.15)	
Well	265 (21.96)	60 (25.00)	144 (22.43)	61 (18.77)	
Very well	22 (1.82)	6 (2.50)	11 (1.71)	5 (1.54)	
Income	13395.11 ± 374.99	13770.86 ± 1243.304	14001.7 ± 432.80	11580 ± 531.54	0.029
Gender					
Male	911 (74.13)	161 (67.08)	528 (79.88)	222 (67.68)	0
Female	318 (25.87)	79 (32.92)	133 (20.12)	106 (32.32)	
Age					
Age < 31	69 (5.60)	15 (6.25)	37 (5.57)	17 (5.17)	0.001
30 < Age < 61	806 (65.37)	176 (73.33)	439 (66.11)	191 (58.05)	
Age > 60	358 (29.03)	49 (20.42)	188 (28.31)	121 (36.78)	
Degree of education					
Unschooled	389 (31.99)	155 (64.58)	130 (20.00)	104 (31.90)	0
Primary school	439 (36.10)	55 (22.92)	267 (41.08)	117 (35.89)	
Middle school and above	388 (31.91)	30 (12.50)	253 (38.92)	105 (32.21)	
Marital status					
Unmarried	150 (12.44)	10 (4.17)	102 (15.91)	38 (11.69)	0
Married and cohabiting	843 (69.90)	202 (84.17)	428 (66.77)	213 (65.54)	
Divorced or widowed	213 (17.66)	28 (11.67)	111 (17.32)	74 (22.77)	
Participation in agricultural industry development					
No	399 (36.84)	52 (21.67)	250 (42.52)	97 (38.04)	0
Yes	684 (63.16)	188 (78.33)	338 (57.48)	158 (61.96)	
Participation in relocation					
No	841 (82.21)	207 (86.25)	441 (79.03)	193 (85.78)	0.014
Yes	182 (17.79)	33 (13.75)	117 (20.97)	32 (14.22)	
participation in employment helping					
No	823 (81.32)	191 (79.58)	459 (81.67)	173 (82.38)	0.712
Yes	189 (18.68)	49 (20.42)	103 (18.33)	37 (17.62)	
Enjoying the health poverty alleviation project					
No	292 (26.43)	72 (30.00)	189 (32.53)	31 (10.92)	0
Yes	813 (73.57)	167 (70.00)	392 (67.47)	253 (89.08)	
Enjoying basic living standard guaranteeing					
No	463 (44.18)	108 (45.00)	310 (54.20)	45 (19.07)	0
Yes	585 (55.82)	132 (55.00)	262 (45.80)	191 (80.93)	

**Table 3 healthcare-08-00256-t003:** Comparison of health status differences among rural poor residents (%).

Health Status	Very Poor	Poor	Medium	Well	Very Well	*p*
Region						
Shanbei	5	12.92	54.58	25	2.5	0
Guanzhong	17.13	33.18	25.55	22.43	1.71	
Shannan	14.77	34.77	30.15	18.77	1.54	
Income						
The poorer	17.28	34.55	30.89	15.71	1.57	0
Medium	16.31	28.37	28.61	24.59	2.13	
The better	9.82	22.32	38.69	27.38	1.79	
Gender						
Male	15.06	29.89	30.34	22.81	1.91	0.070
Female	11.43	28.89	38.73	19.37	1.59	
Age						
Age <31	2.9	7.25	20.29	68.12	1.45	0
30 < Age < 61	11.5	26.8	36.41	22.76	2.53	
Age > 60	22.19	40.35	26.22	10.95	0.29	
Degree of education						
Unschooled	16.88	28.31	39.22	14.29	1.3	0
Primary school	13.43	35.19	31.25	18.29	1.85	
Middle school and above	12.14	24.29	27.65	33.59	2.33	
Marital status						
Unmarried	25.33	27.33	20	27.33	0	0
Married and cohabiting	9.52	28.92	36.75	22.77	2.05	
Divorced or widowed	23.58	32.55	25.94	15.57	2.36	
Participation in agricultural industrial development						
No	26.88	27.89	28.14	15.58	1.51	0
Yes	6.79	27.47	37.37	26.44	1.92	
Participation in relocation						
No	16.07	25.48	35.12	21.55	1.79	0.006
Yes	7.26	34.64	30.73	25.14	2.23	
participation in employment helping						
No	15.45	27.13	34.18	21.41	1.82	0.714
Yes	11.41	27.17	36.41	22.83	2.17	
Enjoying the health poverty alleviation policy						
No	9.93	21.28	42.91	24.47	1.42	0
Yes	16.13	31.28	30.3	20.44	1.85	
Enjoying basic living standard guaranteeing policy						
No	6.49	24.68	39.39	27.92	1.52	0
Yes	21.48	30.41	29.55	16.49	2.06	

**Table 4 healthcare-08-00256-t004:** Concentration index of the health status of the rural poor residents in different regions.

Living Region	Concentration Index	S.E.	95% CI	
			Lower	Upper
Shannxi province	0.0327	0.0065	0.0199	0.0455
Shanbei	0.0068	0.0100	−0.0128	0.0264
Guanzhong	0.0593	0.0092	0.0412	0.0773
Shannan	0.0354	0.0143	0.0074	0.0635

**Table 5 healthcare-08-00256-t005:** Decomposition of health inequity of the rural poor residents in Shannxi province.

Variables	Elasticity Coefficient	Concentration Index	Contribution	Contribution Rate (%)
Income	0.0020	0.4314	0.0009	2.6281
Region (Baseline: Shanbei)				−24.3803
Guanzhong	−0.1000	0.0832	−0.0083	−25.4225
Shannan	−0.0436	−0.0078	0.0003	1.0422
Gender (Baseline: Male)				
Female	0.0023	−0.1117	−0.0003	−0.8101
Age (Baseline: Age < 31)				15.9273
30 < Age < 61	−0.2154	0.0431	−0.0093	−28.4331
Age > 60	−0.1313	−0.1104	0.0145	44.3604
Education (Baseline: Unschooled)				32.5702
Primary school	0.0395	0.0489	0.0019	5.9044
Middle school and above	0.0471	0.1851	0.0087	26.6658
Marital status (Baseline: Unmarried)				17.5964
Married and cohabiting	0.0747	0.0960	0.0072	21.9468
Divorced or widowed	0.0076	−0.1866	−0.0014	−4.3504
Participation in agricultural industrial development (Baseline: No)				
Yes	0.0905	0.0248	0.0022	6.8580
Participation in relocation (Baseline: No)				
Yes	−0.0058	0.2179	−0.0013	−3.8396
participation in transfer employment (Baseline: No)				
Yes	0.0035	0.1700	0.0006	1.7984
Enjoying the health poverty alleviation project (Baseline: No)				
Yes	−0.0596	−0.0588	0.0035	10.7070
Enjoying basic living standard guaranteeing (Baseline: No)				
Yes	−0.0572	−0.1152	0.0066	0.2016
Residual Error				59.2544
Sum			0.0258	100

**Table 6 healthcare-08-00256-t006:** Decomposition of health inequity of the rural poor residents in different regions.

Variables	Shanbei		Guanzhong		Shannan	
	Contribution	Contribution Rate (%)	Contribution	Contribution Rate (%)	Contribution	Contribution Rate (%)
Income	0.0084	114.2978	0.0085	14.2025	0.0057	16.8537
Gender (Baseline: Male)						
Female	−0.0038	−51.6624	0.0008	1.3086	5.728e−06	0.0168
Age (Baseline: Age < 31)		16.9904		16.4410		0.1675
30 < Age < 61	−0.0023	−31.8516	−0.0131	−21.8416	0	0.0963
Age > 60	0.0036	48.8420	0.0229	38.2826	0.0024	0.0712
The degree of education (Baseline: Unschooled)		115.6037		8.1822		26.5094
Primary school	−0.0013	−17.4394	−0.0011	−1.7521	0.0083	24.3873
Middle school and above	0.0098	133.0431	0.0060	9.9343	0.0007	2.1221
Marital status (Baseline: Unmarried)		78.8593		11.9040		8.0099
Married and cohabiting	0.0091	124.0793	0.0135	22.6562	−0.0135	−39.5929
Divorced or widowed	−0.0033	−45.2200	−0.0064	−10.7522	0.0162	47.6028
Participation in agricultural industry development (Baseline: No)						
Yes	−0.0004	−5.9095	0.0068	11.2917	0.0004	1.3103
Participation in relocation (Baseline: No)						
Yes	−0.0089	−121.1139	−0.0008	−1.2910	0.0003	0.9333
Participation in transfer employment (Baseline: No)						
Yes	−0.0001	−1.4937	0.0038	6.3915	0.0016	4.5595
Enjoying the health poverty alleviation project (Baseline: No)						
Yes	−0.0027	−36.6318	0.0031	0.5226	0.0001	0.2426
Enjoying the basic living standards guaranteeing project (Baseline: No)						
Yes	−0.0056	−76.1106	0.0074	12.3163	0.0055	16.0547
Residual Error		32.8293		81.2694		74.6577
Sum	0.0025	100	0.0514	100	0.0277	100

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
