# Peer review of "The Effects of China’s Targeted Poverty Alleviation Policy on the Health and Health Equity of Rural Poor Residents: Evidence from Shaanxi Province"

_healthcare, 2020, doi:10.3390/healthcare8030256_

Round 1

Reviewer 1 Report

Comments to the author(s)

  1. The revised introduction should motivate the empirical context.
  2. Do the data that are used in the estimations contain (survey) weights or not?
  3. The cross-sectional data that have been used in the paper does not allow to estimate causal effects (see also point 5 below).
  4. The models do not control for health endowment.
  5. The revised version of the paper should note that there is empirical research in economics on unemployment and well-being measures using panel data (https://doi.org/10.1002/hec.1361).
  6. The paper should provide some discussion about the external validity of the results that are presented in tables.
  7. The concluding section should state more policy conclusions and discuss avenues for future studies.

Author Response

Dear reviewer,

Thank you for your interest in this paper and your suggestions are of great value in revising the paper. The following is the responses to your suggestions.

Point one: The revised introduction should motivate the empirical context.

Response one: Thank you for your suggestion and we adjusted the content of the introduction, so that we hope that the empirical context will be more clearer.(the line 51-59 in red )

Point two: Do the data that are used in the estimations contain (survey) weights or not?

Response two: The data used in the estimations doesn’t contain weights. Because we randomly selected rural poor residents as the subject of survey, in order to understand the health status of the rural poor residents objectively.

Point three: The cross-sectional data that have been used in the paper does not allow to estimate causal effects (see also point 5 below).

Response three: Yes, I agree with you. Estimating causal effects is a popular method in the social science research now and the panel data is the better material. because of the limitation of the data, this paper mainly discusses the effect of targeted poverty alleviation policy on the health and health equity of rural poor residents based on the analysis of section-cross data and the decomposition of the concentrated index ,which is also the limitation of this research and we indicate this limitation in the discussion section.(line 433-436 in red).

Point four: The models do not control for health endowment.

Response four: We had thought about this question in the process of the research. In the decomposition of the concentration index, health endowment, such as age, sex, degree of education and marital status were controlled in the model.

Point five: The revised version of the paper should note that there is empirical research in economics on unemployment and well-being measures using panel data (https://doi.org/10.1002/hec.1361).

Response five: Yes. We have read the “Unemployment and self-assessed health: evidence from panel data” and agree with that the panel data is a better way to access the external factors on personal well-being or health. Because there was no available we can use, we use the cross-section date in this research. This point is supplemented in the paper (line 117-120 in red).

Point six: The paper should provide some discussion about the external validity of the results that are presented in tables.

Response six: thank you for your this suggestion. And we’d like to provide some discussion about the external validity of the results in the results section (line 212-214 in red).

Point seven: The concluding section should state more policy conclusions and discuss avenues for future studies.

Response seven: This suggestion is so meaningful. We state more policy conclusions and discuss avenues for future studies in conclusions section.(line 453-458 in red; line 465-469 in red)

Reviewer 2 Report

Thank you for an interesting article on the health status and health equity of rural poor residents in China under the implementation of the regional health policy. Some remarks:

The assumption is that "1,233 rural poor residents" were selected for the surveys and "were derived from a questionnaire survey from 12 prefecture-level cities and areas of Shaanxi". Thus, it is not known whether the data come from a larger survey or whether a group (the poor, as already indicated in the abstract) was selected for this particular survey, or the poorer and the less poor among the analysed poor people (as shown in Table 2). The authors say in the abstract that the poor were selected and then divide the studied group into classes (table 2: “very poor” ... “very well" (wealthy??)”).

It is also not fully clear when the study was conducted (2017 or 2007 - see lines 18, 343 and 128).

A simple conversion rate for yuan (into dollars or euros) should be added for easier orientation by the international reader.

There is a lot of disorder in the text - the information is given quite chaotically or need :

The table shows the number of policy measures and later results show that (line 266): “development was significantly better than that of the poor residents who did not participate in industry development”. This is hard to follow what is behind the measure “Industry development” and therefore hard to understand the relation (strong) with health perception by respondents.

So the reader should know a bit more about the policy measures that are essential for the research presented eg. what is industry development?

It is essential for the correct interpretation of the presented results that the nomenclature indicating that the health examination in the presented provinces took place through self-assessment is consistently used. This assumption must be clear from the beginning to the end of the article.

The article is interesting and worth publishing after reviewing and correcting some logic of the presentation of results and clear definitions, as well as improving some unclear English phrases at times.

Good luck.

Author Response

Dear reviewer,

Thank you for your interest in this paper and your suggestions are of great value in revising the paper. The following is the responses to your suggestions.

Point one: The assumption is that "1,233 rural poor residents" were selected for the surveys and "were derived from a questionnaire survey from 12 prefecture-level cities and areas of Shaanxi". Thus, it is not known whether the data come from a larger survey or whether a group (the poor, as already indicated in the abstract) was selected for this particular survey, or the poorer and the less poor among the analyzed poor people (as shown in Table 2). The authors say in the abstract that the poor were selected and then divide the studied group into classes (table 2: “very poor” ... “very well" (wealthy??)”).

Response one: the data come from the rural poor residents who are identified as rural poor residents by government in 2013, according to the standards of targeted poverty alleviation policy. The stands are shown in line 69-71. We have also made a supplementary explanation in the sampling method section (line 128). And then according to the income of the rural residents collected by the social survey in this research, we divided them into classes the very the poor, the poor, the medium, the well, and the very well. I hope our response is clear.

Point two:  It is also not fully clear when the study was conducted (2017 or 2007 - see lines 18, 343 and 128).

Response two: I am sorry to say that there is a mistake in the writing. The data was collected in 2017. I have revised the mistake in the paper.

Point three: A simple conversion rate for yuan (into dollars or euros) should be added for easier orientation by the international reader.

Response three: thank you for your suggestion and we added the conversion rate for yuan (into dollar) in the paper.(line 70, line 227))

Point four: There is a lot of disorder in the text - the information is given quite chaotically or need :The table shows the number of policy measures and later results show that (line 266): “development was significantly better than that of the poor residents who did not participate in industry development”. This is hard to follow what is behind the measure “Industry development” and therefore hard to understand the relation (strong) with health perception by respondents.So the reader should know a bit more about the policy measures that are essential for the research presented eg. what is industry development?

Response four: Thank you for your question. The industry refers to agricultural industry in this research, which mainly includes agricultural cultivation ,livestock breeding, tourism and E-commerce. We revised the express in the tables, the independent variables section, the results section and discussion section.

Point five: It is essential for the correct interpretation of the presented results that the nomenclature indicating that the health examination in the presented provinces took place through self-assessment is consistently used. This assumption must be clear from the beginning to the end of the article.

Response five: I agree with you. In the outcome variables section, we had introduced some methods to measure the health status and self reported-health was used in some research. And introducing that this research use the self-reported health to measure the health status of rural poor residents. So all the health analysis in this paper refers to self-reported health.

Point six: The article is interesting and worth publishing after reviewing and correcting some logic of the presentation of results and clear definitions, as well as improving some unclear English phrases at times.

Response six: Thank you for your interest. We have carried on the careful revision and the expression examination to the article, which made the obviously better than the first manuscript.

This manuscript is a resubmission of an earlier submission. The following is a list of the peer review reports and author responses from that submission.